# Targeting the FABP Axis: Interplay Between Lipid Metabolism, Neuroinflammation, and Neurodegeneration

**DOI:** 10.3390/cells14191502

**Published:** 2025-09-25

**Authors:** Chuantao Wu, Jiejing Lin, Qikai Chen, Wenxue Zhao, Ichiro Kawahata, An Cheng

**Affiliations:** 1School of Medicine, Shenzhen Campus of Sun Yat-sen University, Sun Yat-sen University, Shenzhen 518107, China; wucht3@mail2.sysu.edu.cn (C.W.); linjj86@mail2.sysu.edu.cn (J.L.); chenqk5@mail2.sysu.edu.cn (Q.C.); zhaowx5@mail.sysu.edu.cn (W.Z.); 2Department of Molecular Genetics, Institute of Biomedical Sciences, Fukushima Medical University Graduate School of Medicine, Hikarigaoka, Fukushima 960-1295, Japan; kawahata@tohoku.ac.jp; 3Department of Ophthalmology, School of Medicine, University of California San Francisco (UCSF), San Francisco, CA 94143, USA

**Keywords:** fatty acid-binding proteins (FABPs), metabolic reprogramming, neuroinflammation, neurodegenerative diseases, glial metabolism, FABP-targeted therapy

## Abstract

Fatty acid-binding proteins (FABPs) represent a superfamily of intracellular lipid chaperones essential for the transport of lipids and homeostatic lipid metabolism. Although well-known for their role in systemic metabolic diseases, emerging evidence has identified brain-expressed FABPs as core players in neurodegeneration. This review focuses on brain-expressed FABPs, synthesizing recent advancements that link their role in metabolic dysregulation to neurotoxicity. We present a system that integrates these proteins within a multi-tiered complex pathobiological system that involves: an advanced glial “meta-inflammation” paradigm; a novel view on proteotoxicity via liquid–liquid phase separation (LLPS); changes in the gut–brain axis; and an involvement in the regulation of ferroptosis. Additionally, we also discuss the emerging pharmacological pipeline, highlighting notable preclinical ligands and drawing important lessons from systemic disease first-in-class-targeted FABPs. These first-in-class therapies have successfully validated this target family in systemic diseases. Finally, we explore future therapeutic strategies, where we emphasize the challenges and the precision cell-type-specific delivery approaches to harness the full therapeutic potential of these pivotal proteins.

## 1. Introduction

Over the past decade, the therapeutic strategies for neurodegenerative diseases (NDDs) have mainly focused on downstream pathological hallmarks, such as β-amyloid plaques and neurofibrillary tangles in Alzheimer’s disease (AD) [1]. However, these approaches have consistently failed in clinical trials [2], causing a shift toward investigating upstream causative mechanisms. There is growing evidence that a chronic, low-grade inflammatory state driven by metabolic imbalance (termed “meta-inflammation”) is a core pathogenic process in many NDDs [3]. The primary purpose of this review is to synthesize the burgeoning evidence positioning FABPs as central mediators of this process, thereby establishing them as high-potential therapeutic targets.

Under this new paradigm, neuroinflammation is no longer viewed as a passive response to neuronal death, but rather as a core process actively driven by fundamental changes in cellular metabolic patterns, particularly lipid metabolism. At the core of this metabolic–inflammatory network is the fatty acid binding protein family. Fatty acid binding proteins (FABPs) are a class of cytoplasmic chaperone proteins with molecular weights of approximately 14–15 kDa that specialize in regulating intracellular lipid transport and metabolic responses [4,5]. They are known to play a central role in a variety of metabolic and cardiovascular diseases, including obesity, diabetes and atherosclerosis [6]. Their three-dimensional structures are very similar: a barrel skeleton consisting of 10 antiparallel β-strands folded into a “helix–corner–helix” lid. By virtue of this conserved β-barrel cavity, FABP isoforms bind fatty acids and other small hydrophobic molecules with varying affinities and selectivity.

The therapeutic potential of targeting the FABP family has been successfully demonstrated outside the central nervous system (CNS). FABP4, also known as adipocyte protein 2, is a well-established target in systemic metabolic diseases. FABP4 is highly expressed in adipocytes and macrophages, modulates insulin signaling and promotes lipolysis by activating lipases in adipose tissue [7]. Elevated plasma FABP4 levels correlate with obesity and type 2 diabetes mellitus (T2DM), while genetic knockout or pharmacological inhibition of FABP4 in animal models attenuates weight gain, improves insulin resistance, and reduces atherosclerosis [8,9]. The oral pharmacological inhibition of FABP4 by BMS309403, enhanced insulin sensitivity and suppressed inflammatory signaling in preclinical models, establishing FABP inhibition as a promising strategy for metabolic disorders [4,10,11].

Based on this precedent, research focus has shifted to FABPs expressed in the CNS: FABP3 (cardiac type), FABP5 (tactile type), and FABP7 (brain type) [12]. CNS has a very high lipid content, with lipids accounting for about half of the dry weight of brain tissue, making it the second most lipid-rich organ after adipose tissue, and a very high lipid diversity. Fatty acids and their metabolites maintain brain homeostasis and regulate number of neurological functions, including cell survival, neural neogenesis, and synapse shaping. Brain neuronal cells and glial cells are heavily dependent on transient changes in fatty acid and lipid metabolism [13]. All three FABPs are expressed in the brain and involved in fatty acid metabolism in neuronal cells, providing energy to neuronal cells and maintaining normal function and structure of neuronal cells. In models of Alzheimer’s disease, Parkinson’s disease, and multiple sclerosis, activated microglia and reactive astrocytes show significant Fabp3/5/7 transcriptional up-regulation [14,15,16]. This upregulation is associated with a shift in glial metabolism toward a Warburg-like phenotype, characterized by enhanced glycolysis and accumulation of lipid peroxidation products. Blockade of these FABP isoforms by genetic intervention or brain-permeable small molecule inhibitors reverses the Warburg-like metabolic phenotype, inhibits the NF-κB/NLRP3 inflammasome cascade, and significantly reduces neuronal apoptosis and axonal degeneration [17].

This review will systematically explore the multifaceted roles of FABPs in NDDs. We will begin by dissecting the mechanisms through which FABPs drive metabolic dysregulation in different CNS cell types, including the “meta-inflammatory” axis in glia and the “metabolic survival” balance in oligodendrocytes. We will then connect FABP-mediated lipid dyshomeostasis to proteotoxicity from a liquid–liquid phase separation (LLPS) perspective, explore the emerging gut–brain-FABP axis, and detail the novel link between FABPs and ferroptosis. Subsequently, we will review the current landscape of pharmacological modulators, categorizing them by their CNS penetrance and therapeutic applications. Finally, we will conclude by summarizing the key findings and offering a forward-looking perspective on the challenges and future directions for developing precision FABP-targeted therapies.

### Search Strategy and Selection Criteria

This review was conducted based on a comprehensive literature search of the PubMed, Scopus, and Web of Science databases for articles published up to May 2025. The search strategy employed a combination of keywords and medical subject headings terms, including “Fatty Acid Binding Protein,” “FABP,” “neurodegeneration,” “neuroinflammation,” “Alzheimer’s Disease,” “Parkinson’s Disease,” “Multiple Sclerosis,” “amyotrophic lateral sclerosis,” “glial metabolism,” “microglia,” “astrocyte,” “oligodendrocyte,” “ferroptosis,” “gut–brain axis,” “LLPS,” and the names of specific pharmacological agents such as “BMS309403,” “MF6,” “MF1,” “HY08,” and “ART26.12.” We included peer-reviewed original research articles, meta-analyses, and comprehensive reviews written in English. The selection criteria prioritized studies that elucidated the molecular mechanisms of FABPs in the CNS, their role in the pathophysiology of NDDs, and the development and preclinical validation of FABP-targeted inhibitors.

## 2. FABP-Mediated Metabolic Dysregulation: A Core Driver of Neurodegeneration

### 2.1. The FABP-Driven “Meta-Inflammatory” Axis in Glial Cells

Under physiological conditions, neurons and astrocytes consume most of the glucose in the brain [18]. However, in pathological states, activated microglia and astrocytes undergo metabolic reprogramming [19]. This reprogramming elevates glucose uptake and anaerobic glycolysis and activates the pentose phosphate pathway, while often maintaining mitochondrial function [20]. This metabolic shift supports the high energy and biosynthetic demands of activated glial cells and is tightly coupled to the amplification of inflammatory signaling [21].

FABP5 and FABP7 in astrocytes, and FABP4 in microglia, are key drivers of this “meta-inflammatory” axis. In inflammatory states, these glial cells shift from oxidative phosphorylation to glycolysis. FABP5 and FABP7 in astrocytes capture and deliver peripherally and locally sourced long-chain fatty acids (LCFAs) to provide biosynthetic precursors for rapidly proliferating cells. Concurrently, by delivering fatty acids into the nucleus, they bind to receptors such as PPARs, which deregulates their inhibition of NF-κB, and directly drives the transcription and secretion of pro-inflammatory cytokines like IL-1β, TNF-α, and IL-6 [4,22].

In microglia, FABP4 is now recognized as a core driver of neuroinflammation [23]. Its expression is critical for their inflammatory and metabolic responses to stimuli like lipopolysaccharide (LPS) [24]. The mechanism involves a classic signaling cascade where LPS binds to Toll-like receptor 4 (TLR4), leading to the recruitment of the adaptor protein MyD88 [25,26]. This initiates a downstream cascade involving TRAF6 and TAK1, which activates two major inflammatory pathways: the IKK complex, leading to the phosphorylation and degradation of IκB and subsequent nuclear translocation of NF-κB, and the MAP kinase pathway, leading to JNK phosphorylation and AP-1 activation [27]. Both NF-κB and AP-1 drive the transcription of genes encoding reactive oxygen species (ROS) and inflammatory cytokines. Critically, this process creates a self-amplifying positive feedback loop: NF-κB activation upregulates the expression of FABP4 itself, which in turn enhances TLR4 expression and JNK phosphorylation, thus perpetuating the inflammatory state [28].

Furthermore, FABP4 also acts via a FABP4-UCP2 axis, where FABP4 inhibition leads to an increase in uncoupling protein 2 (UCP2), thereby attenuating ROS production and neuroinflammation [29]. Thus, FABP4 serves as a critical bridge connecting systemic metabolic dysfunction to central neuroinflammation, acting as the molecular entity of “meta-inflammation” within the brain’s resident immune cells (Figure 1).

### 2.2. The FABP-Driven “Metabolic-Survival” Balance in Oligodendrocytes

Myelin, the lipid-rich sheath surrounding neuronal axons, is critical for rapid nerve impulse conduction. Its formation and repair play a crucial role in the normal function of the nervous system, and requires the rapid synthesis of a large number of lipids membranes in a short period of time, involving a variety of key molecules [30], a process heavily dependent on FABP5 and FABP7 [31]. These FABPs facilitate the efficient transport of LCFAs and cholesterol from astrocytes and plasma into oligodendrocytes [32,33].

The two isoforms play distinct but complementary roles in this process. FABP5 preferentially delivers saturated/monounsaturated fatty acids to the mitochondrial matrix. There, they undergo β-oxidation to generate acetyl-CoA, which fuels the TCA cycle and oxidative phosphorylation, thereby generating the massive amounts of ATP and NADPH required to power the energetically demanding process of myelin synthesis and to counteract oxidative stress [33]. In contrast, FABP7 is enriched in the myelin precursor membrane system, where it is directly involved in the assembly of phospholipids and sphingolipids and enhances mitochondrial biogenesis by interacting with PGC-1α to ensure that lipid synthesis is synchronized to match the energy requirements [34]. This division of labor—FABP5 for energy production and FABP7 for biosynthesis and coordination—is critical for efficient myelin formation (Figure 2).

When FABP5/7 expression or function is impaired, fatty acids accumulate abnormally in the cytoplasm and mitochondria, leading to mitochondrial lipid overload, decreased respiratory chain activity, and a collapse of the mitochondrial membrane potential, excess lipoyl-CoA further induces a ROS burst and lipid peroxidation, triggering the opening of the mitochondrial permeability transition pore (mPTP), and ultimately triggering apoptosis or iron death in oligodendrocytes [35]. Animal models revealed that mitochondrial swelling, myelin basic protein (MBP) deficiency, and apoptotic markers (cleaved-caspase3) were significantly increased in mitochondria of Fabp5/7 double-knockout mice in Cuprizone-induced demyelination lesions, whereas exogenous FABP5/7 overexpression or administration of the small-molecule inhibitor MF-6 restored mitochondrial homeostasis, reduced ROS, and promoted myelin regeneration [36]. Thus, FABP5/7 plays a role in the transport and metabolism of fatty acids and provides the necessary lipid raw materials for myelination, thereby influencing myelin formation and development.

### 2.3. The Metabolic Link Between FABPs and Proteotoxicity: A Liquid–Liquid Phase Separation Perspective

The aggregation of misfolded proteins, such as α-synuclein (α-syn), is a central feature of many NDDs like Parkinson’s disease (PD) and Multiple System Atrophy (MSA). α-syn is a 17 kDa protein with three binding structural domains, the N-terminal, hydrophobic, and C-terminal domains, and is enriched in aspartic acid and glutamic acid. α-syn binds to lipids to form a complex and facilitates the binding of α-syn to synaptic membranes that activate platelet-activating factor [37]. Epidemiological evidence of the molecular dynamics of α-syn and in vitro studies have suggested that disorders of lipid homeostatic disturbances may play a role in the pathogenesis of α-syn aggregation [38]. For a long time, FABPs were thought to exacerbate α-syn aggregation by promoting the formation of lipid droplets (LDs), which provide a hydrophobic surface for protein aggregation. However, this description can be deepened with the cutting-edge concept of LLPS, which is now considered a key mechanism driving the formation of membrane less organelles and the initiation of pathological protein aggregation [39,40].

The mechanistic logic is as follows: first, the biogenesis of LDs is itself driven by the LLPS of neutral lipids in the endoplasmic reticulum [41]. Second, and critically, the aggregation of key NDD-related proteins, including α-syn and tau, is now confirmed to be initiated via LLPS [40,42]. In this process, proteins first form liquid-like condensates that subsequently “mature” into irreversible, toxic solid-state aggregates. Therefore, the role of FABPs is far more than providing a passive “hydrophobic surface”. They are very likely active regulators of the LLPS process itself. By precisely controlling the concentration and composition of specific fatty acids within the cell, FABPs directly modulate the thermodynamic environment of the cytoplasm, thereby promoting or inhibiting the phase separation of intrinsically disordered proteins like α-syn. This perspective elevates the role of FABPs from passive “accomplices” to “active regulators” of the first step in proteotoxicity, greatly enhancing their pathological significance.

Beyond generally influencing the lipid environment, specific FABPs can directly participate in the aggregation process [43,44,45]. For instance, recent research has uncovered a direct interaction between FABP7 and α-syn, demonstrating that they form hetero-aggregates which exhibit significantly higher neurotoxicity than α-syn aggregates alone [44]. This finding introduces a novel mechanism where FABP7 acts not just as a lipid transporter but as a direct co-factor in the formation of toxic protein species. Further investigation revealed that these highly toxic hetero-aggregates are selectively propagated into oligodendrocytes and Purkinje cells, the cell types most affected in MSA—via an epsin-2-dependent endocytosis pathway. This work provides a molecular basis for the cell-specific pathology in certain synucleinopathies and establishes FABP7 as a critical mediator linking lipid metabolism directly to the aggregation and cell-to-cell spread of α-syn.

### 2.4. The New Frontier: The Gut–Brain-FABP Axis and Systemic Inflammation

The long-held view of NDDs as confined to the CNS is being challenged by the concept of the “gut–brain axis” [46]. A central tenet of this model is that pathology can originate in the periphery and spread to the brain [47]. In synucleinopathies like PD, compelling evidence suggests that misfolded α-syn first appears in the enteric nervous system (ENS) and propagates to the brainstem via retrograde axonal transport along the vagus nerve in a prion-like fashion [48].

The intestinal protein FABP2 plays a crucial, albeit indirect, role in initiating this cascade. FABP2 is vital for maintaining intestinal barrier integrity by regulating tight junction proteins [49]. In conditions of gut dysbiosis, often driven by poor diet or aging, FABP2 function can be impaired [49,50]. This leads to increased intestinal permeability, a condition known as “leaky gut”. This barrier dysfunction has two critical consequences. First, it is thought to create a local inflammatory environment within the gut that promotes the initial misfolding and aggregation of α-syn in the ENS. Second, it allows bacterial endotoxins, particularly LPS, to leak from the gut lumen into the systemic circulation, causing chronic, low-grade systemic inflammation (Figure 3).

This systemic inflammation represents a “second hit” to the CNS. Circulating inflammatory molecules like LPS can compromise the integrity of the BBB, further promoting neuroinflammation. Once in the brain, LPS directly activates microglia via the FABP4/TLR4 signaling pathway described earlier [24,51], establishing a persistent neuroinflammatory state. This chronic inflammation creates a toxic environment that can exacerbate the neurodegeneration caused by the primary α-syn pathology spreading via the vagus nerve. Thus, the gut–brain-FABP axis represents a multi-faceted pathogenic model where peripheral FABP2 dysfunction initiates events in the gut, leading to both direct prion-like propagation of proteopathy and an indirect, systemic inflammatory response that is mediated centrally by FABP4 [49] (Figure 3). This highlights the utility of peripherally restricted inhibitors, which may succeed not by acting directly on the brain, but by cutting off this peripheral “fuel supply” for central inflammation.

### 2.5. The FABP-Ferroptosis Link: A New Frontier in Neurotoxic Cell Death

Ferroptosis is a form of regulated, iron-dependent cell death driven by the overwhelming accumulation of lipid peroxides, particularly from polyunsaturated fatty acids (PUFAs) within phospholipids [52]. FABP’s ferroptosis connection is novel, but recent research has unequivocally proven this connection.

FABP5 has been posited as a key driver of ferroptosis especially in the context of cerebral hypoxia, stroke and cancer [53,54]. Under cellular stress, FABP5 is upregulated and acts in a positive feedback loop, facilitating the transport and incorporation of PUFAs into cellular membranes. This enriches the membrane with lipids that are highly susceptible to peroxidation, thereby increasing the cell’s vulnerability to ferroptotic death. In contrast, FABP7 has been identified as a protector against ferroptosis [55]. FABP7 actively sequesters vulnerable PUFAs, transporting them away from the cell membrane and into the relative safety of lipid droplets for storage. This mechanism effectively isolates PUFAs from the iron-dependent peroxidative machinery, shielding cells like astrocytes from ferroptotic death. This dichotomous regulation—FABP5 as a sensitizer and FABP7 as a protector—positions the FABP axis as a critical control point in the balance between cell survival and ferroptotic death (Figure 4 and Table 1).

## 3. Pharmacological Modulation of FABPs:

### 3.1. CNS-Penetrant Inhibitors: Resetting Central Metabolism

The clinical application of FABP-targeted therapies is still constrained by the BBB which limits drug access to the CNS and complicates therapeutic delivery. Thus, the design of CNS drugs requires ligands that not only demonstrate high-affinity and isoform-selective FABP binding but also robust BBB permeation [97]. There is now a better ability to overcome these challenges due to advances in medicinal chemistry. These CNS penetrating FABP inhibitors can modulate lipid signaling, glial activation, and the neuroinflammatory processes in NDDs. The following sections discuss preclinical studies of select drugs of this class, and their rationale related to design, pharmacokinetics, as well as preclinical efficacy.

#### 3.1.1. MF6: A CNS-Penetrant Candidate Targeting the Meta-Inflammatory Axis

FABP Ligand 6 (MF6) acts as a selective inhibitor and exhibits high affinity to FABP7 (Kd: 20 nM) [43] and moderate affinity to FABP5 (Kd: 874 nM) [98]. Most notably, its weaker affinity to the cardiac-specific FABP3 (Kd: 1038 nM) [66] decreases the risk for possible cardiotoxic side effects, making it a well-suited candidate for CNS therapies. This favorable selectivity profile allows it to modulate neuroinflammatory pathways while minimizing off-target effects in the periphery.

The therapeutic potential of MF6 is supported by strong preclinical evidence demonstrating both anti-inflammatory and neuroprotective effects. At a cellular level, MF6 demonstrated protective effects on oligodendrocyte precursor cells by stabilizing their mitochondrial membrane potential [80]. In LPS-stimulated primary astrocytes, MF6 provoked a marked reduction in release to pro-inflammatory cytokines IL-1β and TNF-α [84]. These cellular protective benefits translate to clinically meaningful improvements in animal models. In the MOG-induced experimental autoimmune encephalomyelitis (EAE), a model for multiple sclerosis (MS), daily oral MF6 (1 mg/kg for 28 days) significantly delayed disease onset and improved neurological scores alongside reduced spinal cord demyelination and glial activation [84]. Likewise, in the PLP-hα-Syn transgenic mouse model of MSA, treatment with MF6 mitigated cerebellar α-syn inclusions, reduced glial inflammation and oxidative stress, preserved Purkinje neuron morphology, and improved motor coordination [99].

Moreover, MF6 demonstrates CNS drug advantages in pharmacokinetics. Its pharmacokinetics confirmed rapid CNS penetrative capability and therapeutic concentrations following oral dosing. In a model of ischemia and reperfusion, 3 mg/kg administered orally showed significant action with a 30% reduction in infarct volume 30 min post dose highlighting great oral bioavailability and rapid action [100]. This evidence establishes MF6 as a penetrative CNS compound able to influence and modulate glial meta-inflammation and even lessen neurodegenerative pathology in models of MS, MSA and other NDDs (Figure 5).

#### 3.1.2. Targeting FABP3 in Neurotherapeutics: From MF1 to the High-Potency Ligand HY-11-9

MF1 is a pyrazole-type small molecule ligand engineered to selectively target FABP3, possessing a moderate binding affinity of Kd: 302.8 ± 130.3 nM [66]. Its design was based on the changes made to BMS309403 to increase the selectivity for FABP3 while minimizing the unwanted targeting of FABP4 [98]. Preclinical research suggests MF1 could serve as a dual-mechanism therapeutic drug, exhibiting potential both for the treatment of synucleinopathies, such as PD, as well as for seizure disorders, including epilepsy.

The primary mechanism of MF1 in the context of PD is its ability to disrupt pathological protein aggregation [65]. Because FABP3 is highly expressed in dopaminergic neurons and promotes the oligomerization of α-syn, MF1 intervenes by inhibiting this key interaction. This was validated both in vitro, where MF1 prevented α-syn clumping in cells [66], and in vivo in mouse models of PD. In these animal studies, oral administration of MF1 led to significant neuroprotection and functional recovery, including reduced α-syn accumulation, preservation of dopaminergic neurons, and improved motor and cognitive functions [66,101]. These successful outcomes strongly suggest that MF1 has effective CNS penetration and good oral bioavailability.

In addition to its function in neurodegeneration, MF1 has a second, distinct therapeutic action as an anticonvulsant. MF1 has previously shown a considerable degree of seizure suppression in MF1 epilepsy mouse models. This effect is thought to arise from its GABA-A receptor ‘turning on’ action through the benzodiazepine recognition site, a completely different mechanism from its action on FABP3 receptors [102]. While CNS pharmacokinetic data is sparse, pharmacokinetic data from animal models has shown good oral bioavailability and central bioavailability.

Building on this work, HY-11-9, a newly designed and next-generation potent and selective FABP3 ligand [103]. Comparative data show that HY-11-9 has an affinity for FABP3 (Kd≈11.7 nM) that is approximately 25 times higher than that of MF1. More importantly, in a side-by-side comparison made for testing neuroprotection of PD theragnostics in a 1-methyl-4-phenyl-1,2,3,6-tetrahydropyridine (MPTP)-induced PD model, using oral administration of HY-11-9 (0.03 mg/kg), which ameliorated motor deficits, was not observed with MF1 treatment. In further studies, HY-11-9 depletion of pS129-α-syn and reversing MPTP toxicity on dopaminergic neurons was explored. These results establish HY-11-9 as a superior preclinical candidate to MF1, setting a new benchmark for potency and efficacy for FABP3-targeted neuroprotection in PD. While MF1 demonstrated the therapeutic potential of inhibiting the FABP3-α-syn interaction, the development of HY-11-9 provides a more powerful tool and clarifies the path forward for developing novel treatments for PD.

#### 3.1.3. HY08: A FABP3/5 Ligand Targeting Mitochondrial Damage

HY08 is a synthetic small-molecule ligand that selectively inhibits FABP3 and FABP5, which are involved in mitochondria disruption during cerebral ischemia. Hydrophobicity shows high affinity for FABP3 (Kd: 24 ± 7 nM) and moderate affinity for FABP5 (Kd: 410 ± 70 nM), while showing negligible binding to FABP7. By inhibiting FABP- and/or carrier-protein-mediated transport of arachidonic acid HY08 mitigates lipid peroxidation and oxidative stress. In pro-apoptotic signal suppression, pro-apoptotic signal blockade is also observed. With ROTENONE-induced oxidative stress in vitro SH-SY5Y cells demonstrated 4-HNE accumulation suppression, mitochondrial membrane potential preservation, and BAX translocation to mitochondria blockade under pro-oxidative conditions [85].

Pharmacokinetic studies demonstrate and support HY08’s potential for CNS applications, as evidenced in a mouse model of ischemic stroke. Infarct volume was significantly reduced in the stroke model following a single oral dose of 0.3 mg/kg, confirming bioavailability, sufficient therapeutic concentration attainment, and ischemic brain and confirming protective action. These results support its ability to cross the blood–brain barrier and achieve central pharmacological activity following systemic administration [85].

The neuroprotective effect showed in mouse models of transient middle cerebral artery occlusion/reperfusion (tMCAO/R) model. These models received a single oral dose of 0.3 mg/kg 30 min post-reperfusion, significantly reduced infarct volume, improved neurological scores, and increased 7-day survival. Mechanistically, HY08 inhibited the ischemia-induced accumulation of FABP3 and FABP5 in mitochondria, thereby preventing lipid peroxidation (reduced 4-HNE levels) and suppressing BAX-mediated apoptotic signaling. These findings support a role for HY08 in protecting ischemic neurons by disrupting FABP-driven mitochondrial injury pathways [85].

### 3.2. Peripherally Restricted Inhibitors: Modulating Neuro-Immune Crosstalk

The successful development of ART26.12 provides a new approach for FABP-targeted therapy. ART26.12 is a third-generation selective inhibitor of FABP5 from the truxillic acid monoester series It exhibits high affinity for FABP5 (inhibition constant [Ki]: 0.77 ±0.08 μM) with markedly lower affinity for FABP3 (Ki: 71.32 ± 2.40 μM), FABP4 (Ki: > 100 μM), and FABP7 (Ki: 18.99 ± 1.81 μM), as determined by fluorescence displacement assay. By inhibiting FABP5-mediated trafficking of endocannabinoids such as anandamide, ART26.12 reduces their catabolism and enhances local endocannabinoid tone. This indirectly activates CB1 signaling, with potential contributions from CB2, TRPV1, and PPARα receptors involved in metabolic regulation and anti-inflammatory responses. Transcriptomic profiling in the periaqueductal gray revealed upregulation of ion channel and GABAergic synapse-related genes, suggesting central neurophysiological adaptations relevant to pain processing and lipid signaling [104]. These findings support ART26.12 as a mechanistically distinct FABP5-targeting compound that modulates both metabolic and inflammatory pathways implicated in neurodegenerative disease.

ART26.12 demonstrates favorable oral pharmacokinetics, with rapid absorption (Tmax ≈ 0.5 h at 10 mg/kg) and high bioavailability (163% at 300 mg/kg), suggesting nonlinear uptake or enterohepatic recirculation. [104]. Although CNS penetration is limited (brain-to-plasma ratio ≈ 2.3%), the compound reaches therapeutically relevant concentrations in the spinal cord (0.4 ± 0.1 μM), a key region for neuroimmune and metabolic integration in pain modulation [104,105]. ART26.12 is well tolerated in preclinical species, with a no observed adverse effect level (NOAEL) of 1000 mg/kg/day in both rodents and dogs [104].

ART26.12 has shown efficacy across multiple peripheral neuropathy models, including oxaliplatin-induced peripheral neuropathy (OIPN), paclitaxel-induced neuropathy (PIPN), streptozotocin-induced diabetic neuropathy, and cancer-induced bone pain. It dose-dependently reverses mechanical allodynia and cold hyperalgesia, preserves body weight during treatment [104,105]. In diabetic neuropathy models, ART26.12 outperformed duloxetine in reversing pain behaviors at earlier time points [105]. Mechanistically, its analgesic effects are mediated by FABP5-dependent endocannabinoid pathways, involving CB1 and PPARα signaling, both of which are associated with glial regulation and lipid-driven inflammation [104]. These results highlight ART26.12 as a lipid-targeting analgesic with potential utility in diseases involving neuroimmune dysregulation and metabolic dysfunction and strongly support the aforementioned “gut–brain axis” and systemic inflammation theories: even intervening in peripheral metabolic/inflammatory pathways can have a profound impact on the symptoms of the CNS (Table 2).

## 4. Conclusions and Future Perspectives

### 4.1. Summary

This review has emphasized the pivotal position of brain-expressed FABPs as key orchestrators of lipid metabolism, placing them at the nexus of metabolic reprogramming, neuroinflammation, and cell death in NDDs. The evidence provided here supports the conclusion that targeting FABP-mediated metabolic shifts in the brain indicates a primary intervention approach to treating NDDs.

As discussed, meta-inflammatory frameworks in glial cells as well as metabolic survival balance in oligodendrocytes rely heavily on FABP-driven lipid transport. The preclinical successes of the ligands discussed here lend strong support to this hypothesis. Molecules MF6, MF1, and HY08 demonstrate that CNS-penetrant compounds can directly influence critical metabolic pathways to rectify complex demyelinating and proteotoxic as well as ischemic injuries. At the same time, the peripherally restricted inhibitor ART26.12 shows that targeting FABP5 pathways, even outside the CNS, can deeply influence neuroinflammatory processes, underscoring the therapeutic importance of peripheral FABP5 in neuroimmune nexus disordered metabolism.

As a result, disparate FABPs can be regarded as active, multi-faceted therapeutic targets that can be utilized to reshape.

### 4.2. Outlook: From “What to Inhibit” to “Where and How to Modulate” with Precision

The attractive preclinical rationale for FABP inhibitors supports therapeutic avenues that are yet to be explored. Future FABP drug discovery will likely evolve from broad inhibition toward more precise strategies.

FABP drug discovery is likely to evolve beyond broad inhibition toward isoform-specific and cell-type-specific selectivity. Next-generation inhibitors may aim to differentially modulate FABP5 and FABP7 in disease-relevant glial populations, such as microglia, astrocytes, and oligodendrocytes, allowing cell-type-specific control of lipid metabolism and inflammatory responses. A promising direction involves integrating lipidomic data to guide ligand design. Integrating lipidomic profiling into FABP-targeted drug design may allow for the development of inhibitors that selectively block the trafficking of disease-relevant lipid species, offering a more tailored and mechanism-driven approach to therapy. Such approaches could enhance therapeutic efficacy and minimize off-target effects. Moreover, FABP isoforms and their lipid cargoes may serve as biomarkers for early diagnosis, patient stratification, and treatment monitoring.

FABP inhibitors act upstream of multiple pathological processes and are thus well suited for combination with agents that target downstream consequences. For instance, combining a FABP3 inhibitor like MF1, which suppresses α-syn oligomer formation, with agents that target aggregate clearance could provide a synergistic, dual-pronged therapeutic strategy. In demyelinating disorders, using MF6 in combination with a pro-myelinating compound could halt glial damage while promoting remyelination. Likewise, combining a peripherally restricted inhibitor like ART26.12 with a centrally acting analgesic may enable comprehensive pain control by addressing both peripheral and central sensitization. These combinatorial strategies may amplify therapeutic efficacy and broaden the clinical utility of FABP-targeted approaches.

Aberrant Post-Translational Modifications (PTMs) (e.g., phosphorylation, ubiquitination) are a hallmark of NDDs, altering protein function, localization, and aggregation. The function of FABPs themselves may be subject to such regulations. A critical unanswered question is whether FABPs undergo PTMs in NDDs. For example, could phosphorylation of FABP5 alter its binding to VDAC-1, or could ubiquitination of FABP7 target it for degradation, weakening its neuroprotective role in ferroptosis? Investigating the PTMs of FABPs opens a new research field and suggests novel therapeutic targets (e.g., inhibiting a kinase that phosphorylates a FABP), which could offer a more refined and specific strategy than broad ligand inhibition.

In summary, this review highlights the overlooked potential of FABPs as bridges within the networks of lipid metabolism, neuroinflammation, and cell death. Spanning from basic mechanisms to therapeutics, focusing on FABP-centric pathways presents a single and coherent approach to altering neurodegenerative processes. As the field moves forward, the comprehensive therapeutic prospects of FABP-targeted strategies in NDDs will require lipidomic, structural, and precision pharmacological interventions.

## Figures and Tables

**Figure 1 cells-14-01502-f001:**
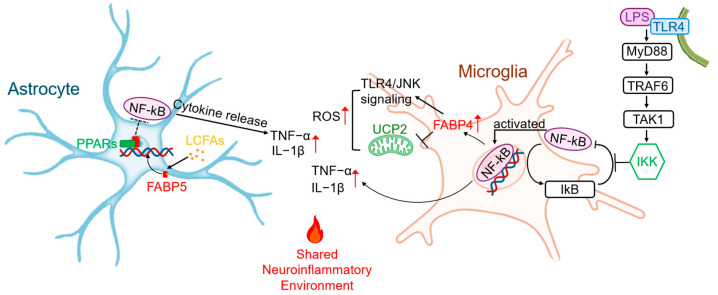
FABP-driven glial “meta-inflammatory” hub. This detailed schematic illustrates the synergistic inflammatory roles of astrocytes and microglia, highlighting specific FABP-mediated signaling pathways. In Microglia: Lipopolysaccharide (LPS) binds to Toll-like receptor 4 (TLR4), recruiting the MyD88 adaptor protein and sequentially activating TRAF6 and TAK1. This initiates the downstream IKK complex, leading to the phosphorylation and degradation of IκB, thereby releasing the NF-κB transcription factor for nuclear translocation. Nuclear NF-κB drives the transcription of pro-inflammatory cytokines (e.g., TNF-α, IL-1β) and, critically, upregulates FABP4 expression. The upregulated FABP4 creates a positive feedback loop that amplifies TLR4/JNK signaling and ROS production while inhibiting the protective UCP2. In Astrocytes: FABP5 transports LCFAs into the nucleus, where they interact with PPARs, de-repressing the inhibition of NF-κB. The activated NF-κB then drives the transcription and release of cytokines such as TNF-α and IL-1β. Together, these glial cells create a Shared Neuroinflammatory Environment that is a key driver of neurodegeneration.

**Figure 2 cells-14-01502-f002:**
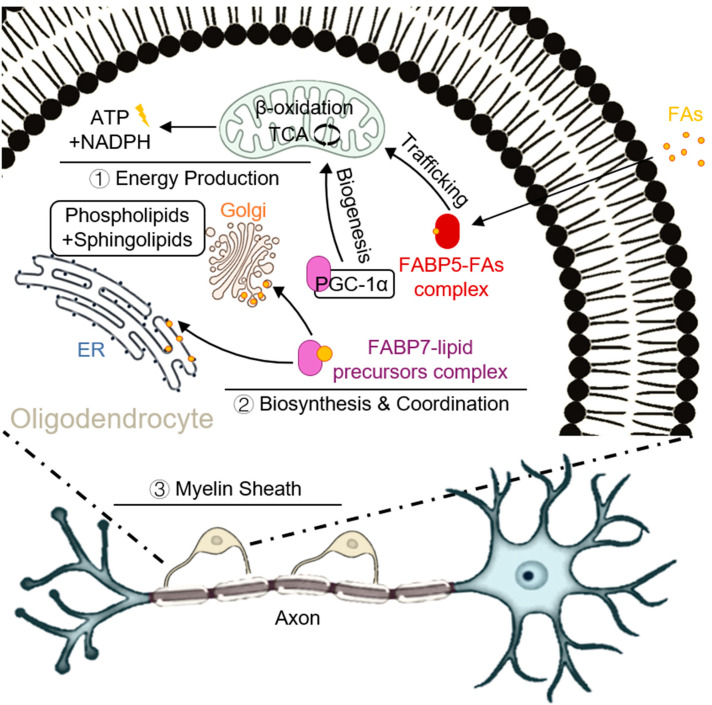
The role of FABP5 and FABP7 in oligodendrocyte lipid metabolism and myelination. This diagram illustrates the complementary functions of FABP5 and FABP7 within a differentiating oligodendrocyte during myelination. External lipids are taken up from the extracellular space. (1) FABP5 Pathway (Energy Production): FABP5 chaperones fatty acids (FAs) to mitochondrion, where they undergo β-oxidation. This process generates acetyl-CoA, which enters the TCA cycle to produce ATP (energy currency) and NADPH (reducing power), both essential for the high energy demands of myelin synthesis. (2) FABP7 Pathway (Biosynthesis and Coordination): FABP7 transports lipid precursors to the endoplasmic reticulum (ER) and Golgi apparatus for the synthesis of core myelin lipids, such as phospholipids and sphingolipids. These lipids are then incorporated into the growing myelin sheath. Additionally, FABP7 interacts with PGC-1α to promote mitochondrial biogenesis, ensuring energy supply matches biosynthetic demand. (3) Final Output: This coordinated action is vital for maintaining oligodendrocyte survival and proper myelin sheath formation around the axon.

**Figure 3 cells-14-01502-f003:**
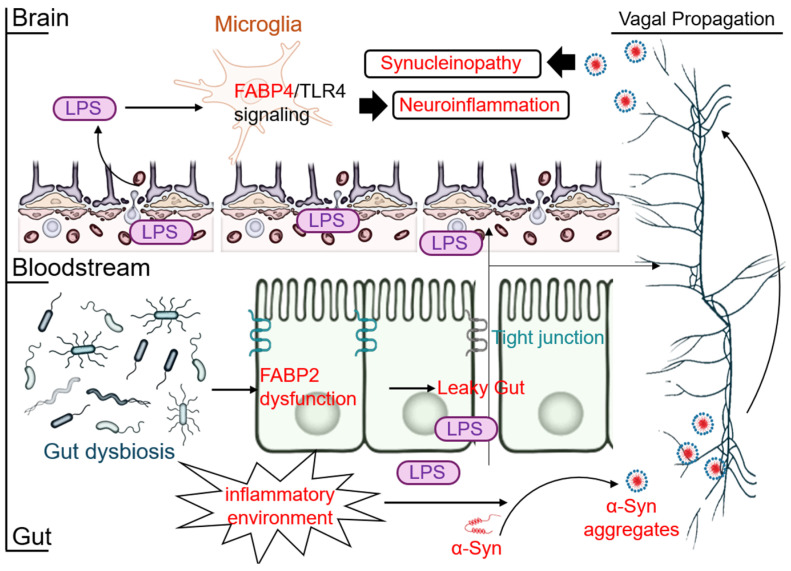
The gut–brain-FABP axis in neurodegeneration. This diagram illustrates the multi-stage pathological cascade linking gut health to neurodegeneration. Stage 1 (Gut Initiation): Gut dysbiosis and other insults lead to FABP2 dysfunction, which compromises tight junctions, resulting in a “leaky gut.” This allows bacterial lipopolysaccharide (LPS) to enter the bloodstream and creates a local inflammatory environment that promotes the misfolding of α-syn in the ENS. Stage 2 (Propagation): The pathology spreads to the brain via two parallel pathways. Pathological α-syn “seeds” propagate from the ENS to the brainstem via retrograde transport along the vagus nerve (Vagal Propagation), consistent with the hypothesis. Concurrently, LPS enters the systemic circulation. Stage 3 (Central Pathology and Inflammation): The α-syn pathology spreads trans-neuronally within the brain, causing synucleinopathy. Meanwhile, circulating LPS crosses a compromised blood–brain barrier (BBB) and activates microglia via the FABP4/TLR4 signaling pathway, establishing a chronic neuroinflammatory state that exacerbates the primary proteotoxic damage.

**Figure 4 cells-14-01502-f004:**
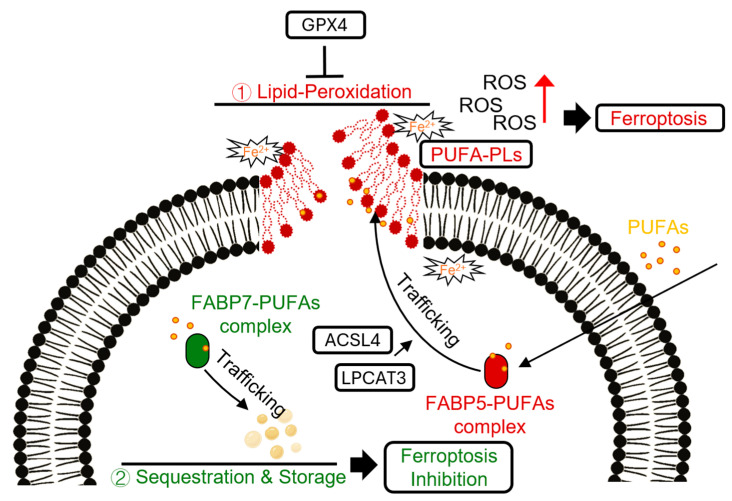
The distinct roles of FABP5 and FABP7 in regulating Ferroptosis. This schematic illustrates the opposing functions of FABP5 and FABP7 in the context of ferroptosis. (1) Pro-ferroptotic pathway (FABP5): FABP5 transports polyunsaturated fatty acids (PUFAs) to the cell membrane, where enzymes like ACSL4 and LPCAT3 incorporate them into phospholipids (PUFA-PLs). In the presence of iron (Fe^2+^), these PUFA-containing phospholipids are highly susceptible to lipid peroxidation, leading to the accumulation of lipid ROS and execution of ferroptosis. This process is normally counteracted by the antioxidant enzyme GPX4. (2) Anti-ferroptotic pathway (FABP7): FABP7 acts as a protective factor by sequestering PUFAs away from the membrane and trafficking them to LDs for safe storage. This prevents their peroxidation and shields the cell from ferroptotic death. The balance between these two pathways can determine the cell’s fate under oxidative stress.

**Figure 5 cells-14-01502-f005:**
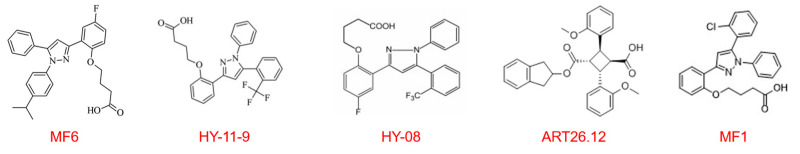
Chemical structures of key preclinical FABP ligands. The 2D chemical structures of representative CNS-penetrant and peripherally restricted FABP inhibitors are shown.

**Table 1 cells-14-01502-t001:** Expression and Roles of Key FABP Isoforms in the Neuro-Gastrointestinal System.

Isoform	Major Peripheral Tissue/Cell Type	Major CNS Tissue/Cell Type	Core Function	Associated Neurodegenerative Disease
FABP2	Small intestine epithelial cells [56]	Not expressed	Fatty acid absorption [57,58], maintaining intestinal barrier integrity [49]	PD (via gut–brain axis) [59]
FABP3	Myocardial cells [60], skeletal muscle cells [61]	Neurons (especially dopaminergic neurons) [62]	Mitochondrial β-oxidation [63,64], promoting α-syn oligomerization [65,66]	PD [65,66], AD [67,68]
FABP4	Adipocytes [69], macrophages [70]	Microglia [24]	Systemic insulin resistance [71], mediating microglial inflammation [72]	Obesity-associated cognitive decline [29], Microglia-mediated neuroinflammation [73]
FABP5	Epidermal cells [74], macrophages [74]	Neurons [75], astrocytes [76,77], oligodendrocytes [78,79,80]	Driving inflammation [81,82,83], regulating myelination [84]	MS [15,84], Stroke [85]
FABP7	Adipocytes [86]	Astrocytes [87,88], radial glial cells [89]	Driving glial meta-inflammation [90,91], neural stem cell development [92,93,94]	MS [84,91,95], MSA [44], AD [87,96], ALS [17]

**Table 2 cells-14-01502-t002:** Preclinical Characteristics of Representative FABP Ligands.

Ligand	Target Isoform and Selectivity	Affinity (Kd/Ki, nM)	CNS Penetrance	Key Preclinical Model	Primary Efficacy Endpoint
MF6 [43,66,84,98]	FABP7 > FABP5 >> FABP3	FABP7: 20; FABP5: 874	Good, Plasma Cmax ~522 nM (4 h)	EAE (MS), MSA	Reduced demyelination, inhibited glial activation, improved motor function
MF1 [66,98,101]	FABP3	302.8	Good (confirmed by in vivo efficacy)	MPTP (PD), Epilepsy	Reduced α-syn aggregation, improved motor and cognitive function, anticonvulsant
HY-11-9	FABP3	11.7	Good (confirmed by in vivo efficacy)	MPTP (PD)	Improved motor function, reduced pS129-α-syn aggregation (superior to MF1)
HY08 [85]	FABP3 > FABP5	FABP3: 24; FABP5: 410	Good (confirmed by in vivo efficacy)	tMCAO/R (Stroke)	Reduced infarct volume, improved neurological function, inhibited mitochondrial damage
ART26.12 [104,106]	FABP5 >> FABP7, FABP3, FABP4	FABP5: 770 (Ki)	Limited (Brain/Plasma Ratio ≈ 2.3%)	Neuropathic Pain	Reversed mechanical allodynia and cold hyperalgesia

## Data Availability

No new data were created or analyzed in this study.

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
