# Peer review of "Targeting the FABP Axis: Interplay Between Lipid Metabolism, Neuroinflammation, and Neurodegeneration"

_cells, 2025, doi:10.3390/cells14191502_

Round 1

Reviewer 1 Report

Comments and Suggestions for Authors

Comments

 Thank you for the opportunity to review this article titled “Targeting the FABP Axis: A Review of the Interplay between Lipid Metabolism, Neuroinflammation, and Neurodegeneration”. The review manuscript is quite interesting, but there are a few inaccuracies that need to be addressed and some areas that could be improved.

  1. The title, “Targeting the FABP Axis: A Review of the Interplay between Lipid Metabolism, Neuroinflammation, and Neurodegeneration.” I think “A Review of…” is redundant in the title; in the authors list, it is missing who is the final author after “and”.
  2. The abstract correctly summarizes the work described in the manuscript. The sentence in the abstract, “This review focuses on FABPs and the recent metabolic dysregulation advancements in neurotoxicity and dysregulation”, purports to explain the authors’ primary purpose in the review but is not clear. Please rephrase.
  3. In the Introduction section, it would be helpful to clearly state the purpose of the review. I suggest the authors to review and edit this section in order to improve its clarity and coherence, and also use this introduction to establish general guidelines for the subsequent sections.
  4. In addition, the literature sources used in the review as well as search strategies and inclusion criteria for publication are not mentioned in the introduction.
  5. The manuscript needs extensive revision for language and grammar.

Minor comments

  1. Line 44: "The therapeutic potential of this family has been successfully demonstrated outside the central nervous system (CNS)”. In this sentence, what do the authors mean by "this family"?
  2. Line-45-46: “FABP4 mainly stored in macrophages and macrophages”: why two macrophages
  3. Figure 1. The illustration lacks clarity and should be improved for better quality and easier comprehension.
  4. For the section 2.2, line 173-211. I suggest adding a figure concept to visually support this section in your review.
  5. Line 443. αSyn oligomer. Missing dash between α and Syn
Comments on the Quality of English Language

n/a

Author Response

[Response to each comment]

We would like to thank the editors and four reviewers for their detailed comments and invaluable suggestions, which helped us to improve the paper significantly. We have carefully addressed all the comments raised by the review team and greatly improved the exposition and presentation of our paper. We mark all the changes in the paper in red to facilitate tracking. We would like to start this response by providing an overview of the major changes we have made in this revision. We then provide detailed point-to-point responses to the review teams’ comments.

Responses to Reviewer #1

Thank you for the opportunity to review this article titled “Targeting the FABP Axis: A Review of the Interplay between Lipid Metabolism, Neuroinflammation, and Neurodegeneration”. The review manuscript is quite interesting, but there are a few inaccuracies that need to be addressed and some areas that could be improved.

  1. The title, “Targeting the FABP Axis: A Review of the Interplay between Lipid Metabolism, Neuroinflammation, and Neurodegeneration.” I think “A Review of…” is redundant in the title; in the authors list, it is missing who is the final author after “and”.

Ans: We thank the reviewer for this excellent suggestion. We revised the title to be more concise and impactful. The new title is: “Targeting the FABP Axis: Interplay between Lipid Metabolism, Neuroinflammation, and Neurodegeneration”. (Page 1, line 2-3)

We apologize for this oversight. The author list has been corrected according to the journal's guidelines to ensure all contributing authors are properly listed. (Page 1, line 4)

  1. The abstract correctly summarizes the work described in the manuscript. The sentence in the abstract, “This review focuses on FABPs and the recent metabolic dysregulation advancements in neurotoxicity and dysregulation”, purports to explain the authors’ primary purpose in the review but is not clear. Please rephrase.

Ans: We agree that the original phrasing was unclear. We have revised the sentence to improve its readability in the abstract section. The revised sentence now reads: “This review focuses on brain-expressed FABPs, synthesizing recent advancements that link their role in metabolic dysregulation to neurotoxicity.” (Page 1, Line 16-18)

  1. In the Introduction section, it would be helpful to clearly state the purpose of the review. I suggest the authors to review and edit this section in order to improve its clarity and coherence and also use this introduction to establish general guidelines for the subsequent sections.

Ans: Thank you for this valuable advice. We have substantially revised the introduction. The purpose of the review is now explicitly stated at the end of the first paragraph (Page 1; Line 38-40), and the final paragraph of the introduction now clearly outlines the structure of the article, providing a roadmap for the reader. (Page 2; Line 82-92)

  1. In addition, the literature sources used in the review as well as search strategies and inclusion criteria for publication are not mentioned in the introduction.

Ans: We acknowledge this was an important omission. As suggested, we have added a new subsection, “1.1. Search Strategy and Selection Criteria,” to the introduction. This section details the methodology, databases, keywords, and inclusion criteria used for our literature search, ensuring transparency and scientific rigor in line with best practices for review articles. (Page 3, Line 93-106)

  1. The manuscript needs extensive revision for language and grammar.

Ans: We fully accept this comment. The manuscript has been thoroughly reviewed and edited by a native English-speaking expert with a background in the relevant academic field to improve clarity, flow, and scientific accuracy.

Minor comments

  1. Line 44: "The therapeutic potential of this family has been successfully demonstrated outside the central nervous system (CNS)”. In this sentence, what do the authors mean by "this family"?

Ans: To improve clarity, we have revised the phrase to: “The therapeutic potential of targeting the FABP family...”. (Page2, Line53 )

  1. Line-45-46: “FABP4 mainly stored in macrophages and macrophages”: why two macrophages

Ans: Thank you for catching this typographical error. We have corrected it to: “FABP4 is highly expressed in adipocytes and macrophages...”. (Page2, Line 55-57)

  1. Figure 1. The illustration lacks clarity and should be improved for better quality and easier comprehension.

Ans: We agree that the original Figure 1 was not sufficiently clear. In line with your feedback and that of Reviewer #3, we have completely redrawn Figure 1. The new figure provides a more detailed and clear illustration of the key signaling pathways involved in glial "meta-inflammation."(Page4,Figure1)

  1. For the section 2.2, line 173-211. I suggest adding a figure concept to visually support this section in your review.

Ans: This is an excellent suggestion. We have added a new figure (New Figure 2) in Section 2.2 to visually explain the distinct yet complementary roles of FABP5 and FABP7 in oligodendrocyte lipid metabolism and myelination. (Page 5, Figure 2)

  1. Line 443. αSyn oligomer. Missing dash between α and Syn

Ans: Thank you for your careful proofreading. We have corrected all instances of "αSyn" to "α-syn" throughout the manuscript for consistency and accuracy.

Reviewer 2 Report

Comments and Suggestions for Authors

Comments to the Author:

  This interesting manuscript by Wu et al. provides a comprehensive overview of data suggesting fatty acid-binding proteins (FABPs) contribute to neurodegeneration through neuroinflammatory mechanism, particularly glial “meta-inflammation,” which is closely linked to the pathological gut-brain axis and ferroptosis. The authors further discuss pharmacological approaches to first-in-class FABP-targeted therapies by summarizing validated pre-clinical ligands and connecting their therapeutic potential and mechanisms of action in neurodegenerative disorders.

   The importance of this review topic is well supported, with numerous references highlighting the role of FABP-associated inflammation in neurodegeneration. Overall, this review offers fundamental and advanced insights, making it accessible to readers unfamiliar with the field while also providing depth for experts. The focus on the “FABP Axis” and “Neurodegeneration” is highly relevant to the scope of the review and should serve as a useful guide for those seeking to learn more about this emerging area.

Minor comment:

   It is recommended to include a schematic diagram illustrating how FABPs induce ferroptosis in Section 2.5 to enhance clarity and reader understanding.

Comments on the Quality of English Language

In line 5, "and" should be removed or replace with a comma.

In lines 46-47, "macrophages" is repeated. Based on the context, it seems you intended to write "adipocytes and macrophages." Please revise accordingly.

In line 47, "promote" should be changed to "promotes" to agree with the singular subject "FABP4".

Additionally, please check the manuscript for English grammar and clarity throughout.

Author Response

[Response to each comment]

We would like to thank the editors and four reviewers for their detailed comments and invaluable suggestions, which helped us to improve the paper significantly. We have carefully addressed all the comments raised by the review team and greatly improved the exposition and presentation of our paper. We mark all the changes in the paper in red to facilitate tracking. We would like to start this response by providing an overview of the major changes we have made in this revision. We then provide detailed point-to-point responses to the review teams’ comments.

Responses to Reviewer #2

Comments to the Author:

This interesting manuscript by Wu et al. provides a comprehensive overview of data suggesting fatty acid-binding proteins (FABPs) contribute to neurodegeneration through neuroinflammatory mechanism, particularly glial “meta-inflammation,” which is closely linked to the pathological gut-brain axis and ferroptosis. The authors further discuss pharmacological approaches to first-in-class FABP-targeted therapies by summarizing validated pre-clinical ligands and connecting their therapeutic potential and mechanisms of action in neurodegenerative disorders.

The importance of this review topic is well supported, with numerous references highlighting the role of FABP-associated inflammation in neurodegeneration. Overall, this review offers fundamental and advanced insights, making it accessible to readers unfamiliar with the field while also providing depth for experts. The focus on the “FABP Axis” and “Neurodegeneration” is highly relevant to the scope of the review and should serve as a useful guide for those seeking to learn more about this emerging area.

Ans: We sincerely thank the reviewer for the positive assessment of the review's importance, breadth, and depth. Your encouragement is greatly appreciated.

  1. It is recommended to include a schematic diagram illustrating how FABPs induce ferroptosis in Section 2.5 to enhance clarity and reader understanding.

Ans: We agree this is an excellent suggestion. To better illustrate the complex relationship between FABPs and ferroptosis, we have added a new schematic diagram (New Figure 4) in Section 2.5. This figure visually distinguishes the opposing roles of FABP5 (pro-ferroptotic) and FABP7 (anti-ferroptotic) in regulating this cell death pathway (Page 8, Figure 4).

Minor comment:

In line 5, "and" should be removed or replaced with a comma. In lines 46-47, "macrophages" is repeated. Based on the context, it seems you intended to write "adipocytes and macrophages." Please revise accordingly. In line 47, "promote" should be changed to "promotes" to agree with the singular subject "FABP4". Additionally, please check the manuscript for English grammar and clarity throughout.

Ans: Thank you for these detailed corrections. We have amended the grammatical errors you pointed out (the "and" in line 5, the repeated "macrophages" in lines 46-47, and "promote" in line 47). Furthermore, the entire manuscript has undergone professional English language editing to ensure clarity and correctness. (Page 1, Line 4; Page 2, Line 55-56)

Reviewer 3 Report

Comments and Suggestions for Authors

In this review, the authors summarize the overlooked potential of FABPs as bridges within the networks of lipid metabolism, neuroinflammation, and cell death. Spanning from basic mechanisms to therapeutics, focusing on FABP-centric pathways presents a single and coherent approach to altering neurodegenerative processes. However, some revisions should be made to show the significance of this study.

  1. The detailed signal pathways that are related to FABP-mediated metabolic dysregulation should be summarized in this review. In addition, a figure that shows the detailed signals should be re-drawn.
  2. The Figure 2 related to the Gut-Brain-FABP Axis in Neurodegeneration should be revised for it is a simple figure, which cannot clearly show the Gut-Brain-FABP Axis in Neurodegeneration. Thus this figure should be revised carefully.
  3. The Figure about the FABP-Ferroptosis Link should be added in this review.
  4. English should be revised by a native English.
Comments on the Quality of English Language

English should be revised.

Author Response

[Response to each comment]

We would like to thank the editors and four reviewers for their detailed comments and invaluable suggestions, which helped us to improve the paper significantly. We have carefully addressed all the comments raised by the review team and greatly improved the exposition and presentation of our paper. We mark all the changes in the paper in red to facilitate tracking. We would like to start this response by providing an overview of the major changes we have made in this revision. We then provide detailed point-to-point responses to the review teams’ comments.

Responses to Reviewer #3

In this review, the authors summarize the overlooked potential of FABPs as bridges within the networks of lipid metabolism, neuroinflammation, and cell death. Spanning from basic mechanisms to therapeutics, focusing on FABP-centric pathways presents a single and coherent approach to altering neurodegenerative processes. However, some revisions should be made to show the significance of this study.

  1. The detailed signal pathways that are related to FABP-mediated metabolic dysregulation should be summarized in this review. In addition, a figure that shows the detailed signals should be re-drawn.

Ans: We thank the reviewer for this important point. To provide a more in-depth discussion of FABP-mediated metabolic dysregulation, we have significantly expanded the relevant sections. Specifically, in Section 2.1 and the revised Figure 1, we now describe the TLR4/MyD88/NF-κB and JNK signaling cascades in microglia in greater detail. (Page4, Figure 1)

New section 2.1 The FABP-Driven "Meta-inflammatory" Axis in Glial Cells

Under physiological conditions, neurons and astrocytes consume most of the glucose in the brain [18]. However, in pathological states, activated microglia and astrocytes undergo metabolic reprogramming [19]. This reprogramming elevates glucose uptake and anaerobic glycolysis and activates the pentose phosphate pathway, while often maintaining mitochondrial function [20]. This metabolic shift supports the high energy and biosynthetic demands of activated glial cells and is tightly coupled to the amplification of inflammatory signaling [21].

FABP5 and FABP7 in astrocytes, and FABP4 in microglia, are key drivers of this "me-ta-inflammatory" axis. In inflammatory states, these glial cells shift from oxidative phos-phorylation to glycolysis. FABP5 and FABP7 in astrocytes capture and deliver peripherally and locally sourced long-chain fatty acids (LCFAs) to provide biosynthetic precursors for rapidly proliferating cells. Concurrently, by delivering fatty acids into the nucleus, they bind to receptors such as PPARs, which deregulates their inhibition of NF-κB, and directly drives the transcription and secretion of pro-inflammatory cytokines like IL-1β, TNF-α, and IL-6 [4, 22].

In microglia, FABP4 is now recognized as a core driver of neuroinflammation [23]. Its expression is critical for their inflammatory and metabolic responses to stimuli like lipo-polysaccharide (LPS) [24]. The mechanism involves a classic signaling cascade where LPS binds to Toll-like receptor 4 (TLR4), leading to the recruitment of the adaptor protein MyD88[25, 26]. This initiates a downstream cascade involving TRAF6 and TAK1, which activates two major inflammatory pathways: the IKK complex, leading to the phosphorylation and degradation of IκB and subsequent nuclear translocation of NF-κB, and the MAP kinase pathway, leading to JNK phosphorylation and AP-1 activation [27]. Both NF-κB and AP-1 drive the transcription of genes encoding ROS and inflammatory cyto-kines. Critically, this process creates a self-amplifying positive feedback loop: NF-κB activation upregulates the expression of FABP4 itself, which in turn enhances TLR4 expression and JNK phosphorylation, thus perpetuating the inflammatory state [28].   

Furthermore, FABP4 also acts via a FABP4-UCP2 axis, where FABP4 inhibition leads to an increase in uncoupling protein 2 (UCP2), thereby attenuating ROS production and neuroinflammation [29]. Thus, FABP4 serves as a critical bridge connecting systemic met-abolic dysfunction to central neuroinflammation, acting as the molecular entity of "me-ta-inflammation" within the brain's resident immune cells (Figure 1).

  1. The Figure 2 related to the Gut-Brain-FABP Axis in Neurodegeneration should be revised for it is a simple figure, which cannot clearly show the Gut-Brain-FABP Axis in Neurodegeneration. Thus, this figure should be revised carefully.

Ans: We agree with advice. We have therefore completely reconceptualized and redrawn this figure (now renumbered as Figure 3). The new figure accurately depicts the current mainstream view of α-synuclein pathology spreading from the gut to the brain primarily via the vagus nerve, incorporating the latest research findings. (Page 6, Line 213- Page 8, line 240)

  1. The Figure about the FABP-Ferroptosis Link should be added in this review.

Ans: As suggested, we have added a new schematic (New Figure 4) in Section 2.5 to illustrate the opposing roles of FABP5 and FABP7 in the regulation of ferroptosis, which we believe will greatly enhance the reader's understanding of this emerging topic (Page 8, Figure 4).

  1. English should be revised by a native English.

Ans: We have taken this comment very seriously. The manuscript has been professionally edited by a native English speaker with expertise in neuroscience to ensure the language is precise, fluent, and meets the highest publication standards.

Reviewer 4 Report

Comments and Suggestions for Authors

This manuscript provides a fairly comprehensive review of the literature on the role of FABP in neuroinflammation and neurodegenerative disorders.  The subject matter is quite interesting, and the paper should be of interest to the readership of Cells.  However, there are a number of corrections that must be made prior to acceptance for publication.  These are listed below.

  1. The name of the final author is missing in the list of authors.
  2. Figure 2 doesn't make sense.  The current think of the alpha synuclein gut-brain axis is that aSyn travels to the brain via the vagus nerve and enters through the brain stem.  What is the evidence aSyn enters the brain via blood?
  3. The authors describe a number of compounds targeting FABP (including describing their chemical class) but do not show their structures. The structure of each compound mention in the article should be shown in a figure.
  4. Page 12, line 445: an aSyn aggregation inhibitor is expected to prevent aggregate formation, not increase aggregate clearance.
  5. Page 12, lines 450 - 458: the discussion in this section is speculation and should be deleted.  

Author Response

[Response to each comment]

We would like to thank the editors and four reviewers for their detailed comments and invaluable suggestions, which helped us to improve the paper significantly. We have carefully addressed all the comments raised by the review team and greatly improved the exposition and presentation of our paper. We mark all the changes in the paper in red to facilitate tracking. We would like to start this response by providing an overview of the major changes we have made in this revision. We then provide detailed point-to-point responses to the review teams’ comments.

Responses to Reviewer #4

This manuscript provides a fairly comprehensive review of the literature on the role of FABP in neuroinflammation and neurodegenerative disorders.  The subject matter is quite interesting, and the paper should be of interest to the readership of Cells.  However, there are a number of corrections that must be made prior to acceptance for publication.  These are listed below.

  1. The name of the final author is missing in the list of authors.

Ans: We apologize for this oversight. The author list has been checked and corrected to ensure it is complete and accurate. (Page 1, line 4)

  1. Figure 2 doesn't make sense. The current think of the alpha synuclein gut-brain axis is that aSyn travels to the brain via the vagus nerve and enters through the brain stem. What is the evidence aSyn enters the brain via blood?

Ans: We are very grateful to the reviewer for raising this critical scientific point. The depiction in our original figure did not accurately reflect the current consensus in the field. This was a significant error, which we have now fundamentally corrected. We have completely rewritten the corresponding section (2.4) and redesigned the figure (now Figure 3). The new text and figure are based on substantial evidence indicating that α-synuclein propagates in a prion-like manner from the enteric nervous system (ENS) to the brainstem via retrograde axonal transport along the vagus nerve, which is the predominantly accepted pathway. (Page 6, Line 213- Page 8, line 240)

New section 2.4. The New Frontier: The Gut-Brain-FABP Axis and Systemic Inflammation

The long-held view of NDDs as confined to the central nervous system is being challenged by the concept of the "gut-brain axis"[46]. A central tenet of this model is that pathology can originate in the periphery and spread to the brain [47]. In synucleinopathies like PD, compelling evidence suggests that misfolded α-syn first appears in the enteric nervous system (ENS) and propagates to the brainstem via retrograde axonal transport along the vagus nerve in a prion-like fashion [48].

The intestinal protein FABP2 plays a crucial, albeit indirect, role in initiating this cascade. FABP2 is vital for maintaining intestinal barrier integrity by regulating tight junction proteins[49]. In conditions of gut dysbiosis, often driven by poor diet or aging, FABP2 function can be impaired [49, 50]. This leads to increased intestinal permeability, a condition known as "leaky gut". This barrier dysfunction has two critical consequences. First, it is thought to create a local inflammatory environment within the gut that pro-motes the initial misfolding and aggregation of α-syn in the ENS. Second, it allows bacterial endotoxins, particularly LPS, to leak from the gut lumen into the systemic circulation, causing chronic, low-grade systemic inflammation (Figure 3).  

This systemic inflammation represents a "second hit" to the CNS. Circulating inflammatory molecules like LPS can compromise the integrity of the blood-brain barrier (BBB), further promoting neuroinflammation. Once in the brain, LPS directly activates microglia via the FABP4/TLR4 signaling pathway described earlier [24, 51], establishing a persistent neuroinflammatory state. This chronic inflammation creates a toxic environment that can exacerbate the neurodegeneration caused by the primary α-syn pathology spreading via the vagus nerve. Thus, the gut-brain-FABP axis represents a multi-faceted pathogenic model where peripheral FABP2 dysfunction initiates events in the gut, leading to both direct prion-like propagation of proteopathy and an indirect, systemic inflammatory response that is mediated centrally by FABP4 [49] (Figure 3). This highlights the utility of peripherally-restricted inhibitors, which may succeed not by acting directly on the brain, but by cutting off this peripheral "fuel supply" for central inflammation.

  1. The authors describe a number of compounds targeting FABP (including describing their chemical class) but do not show their structures. The structure of each compound mention in the article should be shown in a figure.

Ans: We fully agree that displaying the chemical structures is essential for a review discussing pharmacological interventions. We have added a new Figure 5 that clearly shows the chemical structures of MF6, MF1, HY08, HY-11-9 and ART26.12. We have made a note of this in the legend for Figure 5. (Page 10)

  1. Page 12, line 445: an aSyn aggregation inhibitor is expected to prevent aggregate formation, not increase aggregate clearance.

Ans: Thank you for this sharp observation. We acknowledge that our original wording was imprecise. We have revised the sentence to be more accurate: “For instance, combining a FABP3 inhibitor like MF1, which suppresses α-syn oligomer formation, with agents that target aggregate clearance could provide a synergistic, dual-pronged therapeutic strategy.” (Page 13, Line 431-434)

  1. Page 12, lines 450 - 458: the discussion in this section is speculation and should be deleted.

Ans: We deleted this section.

Round 2

Reviewer 1 Report

Comments and Suggestions for Authors

Too many lines in Figures 1 and 2. The quality and visibility could be improved.  

Author Response

Thank you for coordinating the second round of reviews for our manuscript. We are sincerely grateful for the time and effort all reviewers have dedicated to improving our work. We would like to express our profound appreciation to Reviewers #2, #3, and #4 for their positive feedback and for deeming the manuscript suitable for publication. We are also very thankful to Reviewer #1 for the final constructive suggestion to enhance the quality of our figures. We have carefully addressed this final point to polish the manuscript. Below are our point-by-point responses.

Responses to Reviewer #1

Too many lines in Figures 1 and 2. The quality and visibility could be improved."

Ans: We sincerely thank the reviewer for this valuable and constructive feedback; we have professionally redesigned both figures (Page 4; Page 6).

Reviewer 3 Report

Comments and Suggestions for Authors

The revision is OK.

Author Response

Thank you for coordinating the second round of reviews for our manuscript. We are sincerely grateful for the time and effort all reviewers have dedicated to improving our work. We would like to express our profound appreciation to Reviewers #2, #3, and #4 for their positive feedback and for deeming the manuscript suitable for publication. We are also very thankful to Reviewer #1 for the final constructive suggestion to enhance the quality of our figures. We have carefully addressed this final point to polish the manuscript. Below are our point-by-point responses.

Responses to Reviewer #3

The revision is OK.

Ans: We are grateful for the reviewer's positive feedback and for their approval of our revisions.

Reviewer 4 Report

Comments and Suggestions for Authors

The authors have done a good job in responding to the comments raised in the previous review.  This paper is now suitable for publication.

Author Response

Thank you for coordinating the second round of reviews for our manuscript. We are sincerely grateful for the time and effort all reviewers have dedicated to improving our work. We would like to express our profound appreciation to Reviewers #2, #3, and #4 for their positive feedback and for deeming the manuscript suitable for publication. We are also very thankful to Reviewer #1 for the final constructive suggestion to enhance the quality of our figures. We have carefully addressed this final point to polish the manuscript. Below are our point-by-point responses.

Responses to Reviewer #4

The authors have done a good job in responding to the comments raised in the previous review. This paper is now suitable for publication.

Ans: We are sincerely grateful for the reviewer's encouraging comments and endorsement of our manuscript for publication. Your positive assessment is a great encouragement to us.
